# Modeling and Simulating an Orthodontic System Using Virtual Methods

**DOI:** 10.3390/diagnostics12051296

**Published:** 2022-05-23

**Authors:** Stelian-Mihai-Sever Petrescu, Mihaela Jana Țuculină, Dragoș Laurențiu Popa, Alina Duță, Alex Ioan Sălan, Ruxandra Voinea Georgescu, Oana Andreea Diaconu, Adina Andreea Turcu, Horia Mocanu, Andreea Gabriela Nicola, Ionela Teodora Dascălu

**Affiliations:** 1Department of Orthodontics, Faculty of Dental Medicine, University of Medicine and Pharmacy of Craiova, 200349 Craiova, Romania; mihaipetrescu2702@gmail.com (S.-M.-S.P.); marceldascalu@yahoo.com (I.T.D.); 2Department of Endodontics, Faculty of Dental Medicine, University of Medicine and Pharmacy of Craiova, 200349 Craiova, Romania; oanamihailescu76@yahoo.com; 3Department of Automotive, Transportation and Industrial Engineering, Faculty of Mechanics, University of Craiova, 200478 Craiova, Romania; alina.duta@edu.ucv.ro; 4Department of Oral and Maxillofacial Surgery, Faculty of Dental Medicine, University of Medicine and Pharmacy of Craiova, 200349 Craiova, Romania; alexsalan87@gmail.com; 5Faculty of Dental Medicine, University Titu Maiorescu of Bucharest, 67A Gheorghe Petrascu Str., 031593 Bucharest, Romania; ruxi0372@yahoo.com (R.V.G.); horia.mocanu@prof.utm.ro (H.M.); 6Department of Oro-Dental Prevention, Faculty of Dental Medicine, University of Medicine and Pharmacy of Craiova, 200349 Craiova, Romania; adinaturcu14@yahoo.com (A.A.T.); andreea_anghel@yahoo.com (A.G.N.)

**Keywords:** cone beam computed tomography, finite element method, virtual model, elastic forces, result maps, orthodontics

## Abstract

Cone beam computed tomography (CBCT) is a modern imaging technique that uses X-rays to investigate the structures of the dento-maxillary apparatus and obtain detailed images of those structures. The aim of this study was to determine a functional mathematical model able to evaluate the elastic force intensity on each bracket and tube type element and the ways in which those components act on the orthodontic system being used. To analyze a real orthodontic system, we studied the case of a 13-year-old female patient. To transfer geometric information from tomographic images, we used the InVesalius software. This software can generate three-dimensional reconstructions based on sequences and files in the DICOM format and was purchased from CBCT equipment. We analyzed and processed the geometries of the converted tissues in InVesalius using the Geomagic software. After using the Geomagic software, we exported the resulting model to the SolidWorks software used in computer-aided design. In this software, the model is transformed into a virtual solid. After making the geometric model, we analyzed the model using the Ansys Workbench software, which incorporates finite element analysis techniques. Following the simulations, we obtained result maps, which showed the complete mechanical behavior of the analyzed structures.

## 1. Introduction

Numerous etiological factors can affect craniofacial development, leading to the appearance of malocclusions [1]. The growing variety of the clinical aspects of malocclusions has led to the emergence of new methods for diagnosis and treatment.

Cone beam computed tomography (CBCT) is the most promising diagnostic imaging method in recent years and is able to provide images with submillimeter resolutions (2 pairs of lines/millimeter) of a high quality to establish a diagnosis with shorter scan times (approximately 60 s). The radiation dose to which a patient is exposed in a CBCT scan is almost 10 times lower than that in conventional computed tomography of the maxillary bones (68 microsieverts compared to 600 microsieverts); this method also has a higher dimensional accuracy (an increase of about 2%) [2,3,4].

One of the most important features of CBCT is its ability to construct different images, such as panoramic illustrations of adjacent teeth and structures and cephalometric illustrations.

Because of the need to replace expensive and time-consuming methods (3D stereophotogrammetry and standardized facial photographs), a simple alternative using CBCT has emerged. This method provides 3D scans of both soft and hard tissues for diagnosis and treatment planning. A previous study showed that wrapped CBCT images from non-standardized random frontal photographs can be used to analyze soft tissue facial profile measurements [5].

If a major scan is performed, visualizations can generally be done without creating additional two-dimensional panoramic and cephalometric radiographs. These images can be reconstructed from the volume of computed tomography, but it is imperative to include all regions of interest. Several studies have confirmed that the cephalometric images synthesized from the CBCT volume are equivalent to conventional radiography in terms of mark identification, analysis, and overall diagnostic value [6,7,8].

Other acquisition and planning software has been proposed for full orthodontic treatment or orthognathic surgery, including the Total Face Approach (TFA) 3D cephalometric analysis system. This system identifies new evaluation algorithms on the vertical and sagittal planes to determine the symmetry of the patient and create a new classification based on 3D data [9].

The finite element method (FEM) is a modern numerical method of analysis with fields of applicability in dentistry and, implicitly, orthodontics. FEM starts with a real phenomenon that is divided into a number of elements with simple shapes, which are then assembled to build the final geometry of the structure. The principle of this method involves dividing the domain under analysis into a number of subdomains, which are simple geometric elements called “finite elements”, linked together by “nodes”. The FEM equations of a structure consisting of a finite number of discrete elements will form systems of linear equations whose solutions represent the unknowns of the problem such as displacement, strain, and stress [10,11,12].

The aim of this study was to determine the elastic forces that appear during fixed orthodontic treatment and the mechanical effects produced by those forces. A clinical case was considered as an example. For this, we used CBCT images of a patient to develop a 3D model using reverse engineering techniques. Using this model, as well as the metallic elements of the orthodontic system, the forces acting on each tooth were obtained. Then, a simulation was obtained by the finite element method, as well as the specific result maps. This study is novel because we obtained a personalized 3D model of an orthodontic system of a patient starting from CBCT images. Additionally, the system of forces and the results obtained through simulations with the finite element method provides a personalized system for specific orthodontic treatment.

## 2. Materials and Methods

The present study was approved by the Ethics Committee of the University of Medicine and Pharmacy of Craiova, Romania (approval reference no. 72/07.09.2020), in accordance with the ethical guidelines for research with human participants of the University of Medicine and Pharmacy of Craiova, Romania. Written informed consent was obtained from the legal guardian of the subject involved in the study.

To analyze a real orthodontic system, we studied the case of a 13-year-old female patient who presented to the Orthodontic Clinic of the Faculty of Dentistry at the University of Medicine and Pharmacy of Craiova.

Front, left, and right profile photos; intraoral photos in occlusion; and photos from the time of 1.3 surgical exposure and extraction for orthodontic purposes of 1.4 were taken. Figure 1 and Figure 2 show images of the patient.

After the alginate impression of the patient’s dental arches, class III plaster models were cast. Figure 3 shows images of these models.

The patient was also analyzed using CBCT investigations. Figure 4 shows representative images of the maxillary.

After clinical and paraclinical examinations, the patient was diagnosed with angle class I malocclusion, dento-alveolar disharmony with crowding and impaction of 1.3. We applied orthodontic treatment with a fixed device using the straight-wire technique, which is a widely used technique in orthodontics that uses pre-angled brackets. These brackets ensure that all dental movements are made with the fixed orthodontics. The straight wire slides through the slots of the brackets, with the teeth aligning under the effect of elastomeric patterns and directional forces.

After bonding the fixed orthodontic appliance, surgical exposure of 1.3 and orthodontic extractions of 1.4 and 4.4 were performed at the Oral and Maxillofacial Surgery Clinic of the Emergency County Clinical Hospital of Craiova. After performing perimetry, we found a large deficit of space in the maxillary arch (−7 mm) and mandibular arch (−6 mm). Thus, to align the teeth, we resorted to the extraction method.

To obtain the three-dimensional geometry of the patient’s maxillary bones, we used a Capture scanner (3D Systems, Rock Hill, SC, USA) to scan the models. To obtain tomographic images, we used a CS 8200 3D CT scanner (Carestream Dental, Atlanta, GA, USA).

For three-dimensional modeling of the orthodontic metallic element, we analyzed a Sentalloy upper orthodontic wire (Dentsply Sirona, Charlotte, NC, USA) with a diameter of 0.014 inches, a Sentalloy lower orthodontic wire with a diameter of 0.014 inches, and a set of LEGEND Medium brackets and tube components. Visual analysis of the tomographic images was performed using the Syngo FastView medical imaging software (Siemens, Munich, Germany).

Transformation of the different tissues from the tomographic images into three-dimensional geometries of the point-cloud type was done using the InVesalius software (CTI, Campinas, Brazil). This software is used for specialized research because it transforms CT images into primary geometries of the tissues analyzed by medical imaging methods.

The transformation, analysis, and processing of the geometries of the converted tissues from InVesalius were achieved using the Geomagic software (Morrisville, NC, USA), which is specialized in three-dimensional scanning and applies reverse engineering methods.

Subsequently, we exported the model to the SolidWorks software (SolidWorks Corp., Waltham, MA, USA) used in computer-aided design (CAD) and virtual prototyping. In this software, we turned the model into a virtual solid. We also modeled the two orthodontic wires, as well as the bracket and tube elements.

After creating the geometric model, we performed an analysis using the Ansys Workbench software (Ansys Inc., Canonsburg, PA, USA), which incorporates finite element analysis techniques. Following the simulations, we obtained result maps showing the complete mechanical behavior of the analyzed structures.

For simple calculations and to create graphs and diagrams, we used Microsoft Excel. For complicated calculations, we used the Maple software (Maplesoft, Waterloo, ON, Canada). Data storage and functional values were determined using Microsoft Excel. To determine the unknowns from systems of equations with a degree of 6 and 6 unknowns, Maple was used with implementation of the matrix calculations and determinants. Initially, Microsoft Excel was used for this process, but the results were not correct.

Several methods were used to carry out this study, some classical and others innovative, including medical imaging methods, reverse engineering and three-dimensional scanning methods, CAD-type methods, material strength methods, Elasticity Theory methods, and FEM.

To obtain a virtual model of the maxillary, starting from the study model, we used a Capture scanner. This device creates a point cloud model by overlaying several successive scans using another automatic or manual alignment algorithm. Thus, the plaster model was successively rotated manually by about 10°, and at each rotation, we obtained a point cloud.

To obtain the final model, 30 successive scans were performed. Then, we obtained the final model of the scan operation. Figure 5 shows the common point cloud of the upper and lower images, which represents the raw, unprocessed model of the scanned maxillary.

Finally, we obtained a closed final surface, as shown in Figure 6.

Starting from this surface, we intended to obtain a virtual solid. For this purpose, it was necessary that the surface model did not contain self-intersecting areas, common edges, very deformed edges, sharp-spike-type elements, small artifact components, small channels, or holes. Using various Geomagic-specific procedures and reverse engineering, these geometric problems were automatically eliminated (Mesh Doctor command) in two steps, the first involving the identification operation and the second involving the removal operation.

After correcting the geometry, we obtained a closed surface that contained a number of approximately 646,000 elementary triangular surfaces. The CAD software allows transformation of a closed surface into a virtual solid only if the number of triangular elementary surfaces is less than 150,000. To obtain such an area, we used a decimation operation, which reduced the number of elementary areas to about 145,000. Through such operations, we reduced the quality of the surface, so this closed surface also underwent a finishing operation. Finally, the model was exported to the SolidWorks software, where a virtual solid was produced, as shown in Figure 7.

We applied the same process to the mandibular virtual model. This model was also exported to SolidWorks where we turned it into a virtual solid.

We did not use these models directly in the simulation, but certain elements served as a basis for the present study and analysis. A disadvantage of these models was that the complete structure of the teeth and osseous components was not obtained, but these virtual models could be used in analyses that also consider soft tissues. These models are also used for the virtual alignment of teeth.

To obtain the geometry, we used CBCT images realized before gluing the orthodontic appliance. We used the InVesalius software, which creates three-dimensional geometric structures based on the shades of gray in an image by choosing different tissues from the software database. Thus, by choosing bone-type tissues from the software menu, the primary geometries for such structures can be obtained.

We also selected enamel structures from the software menu and obtained the primary geometry for this type of tissue.

These primary geometries containing bone and enamel structures were coupled, and an. stl file was exported to the Geomagic software. This file contained a point cloud similar to that produced in three-dimensional scanning. In this application, the geometric structure was initially transformed into primary triangular surfaces. For this reason, these structures were edited using reverse engineering techniques and methods in the Geomagic software. Figure 8 presents this initial model.

From this initial model, which was considered as a foundation, we first extracted the osseous structure of the maxillary and mandible followed by the dental structure. To obtain models of the osseous components, we successively eliminated, by direct selection, the artifact-type elements and teeth from the model. Initially, the model had 4,863,566 triangular elementary surfaces.

The final model of the osseous components was constructed in Geomagic and then in SolidWorks, where it was automatically transformed into a virtual solid (see Figure 9).

To obtain the dental structure, we used the InVesalius software with the Enamel filter.

From the InVesalius software, the point cloud structure was exported to the Geomagic software for processing and transformation. In this way, we transformed the model into triangular surfaces, initially numbering 3,426,848.

We then processed the model using operations to remove the geometries of the osseous components, decimation operations, operations to remove non-compliant surfaces, etc. After applying these operations, the model had 1,695,551 elementary surfaces.

Finally, we applied other reverse engineering operations to the model. The final model of the dental structure containing 156,042 elementary triangular surfaces is presented in Figure 10.

Finally, we imported the model in SolidWorks, where we transformed it into independent virtual solids. To obtain the complete model, we loaded the two models (osseous and dental) into the Assembly module of SolidWorks. Given that these models have a common global coordinate system because they come from the same set of CBCT images, we aligned the main planes that define the coordinate system and obtained the model shown in Figure 11.

To avoid interference between the virtual solids of the osseous components, we extracted the volumes of the teeth using CAD commands and techniques; these cavities represented the dental alveoli.

To obtain the models of the bracket and tube elements, we used well-known CAD methods and techniques. Initially, we scanned the two-dimensional box with the bracket and tube elements. We then scaled the image to a natural scale, enabling a series of measurements of the image to be made. We also used a digital caliper to measure different sizes.

Using the data obtained by the measurements and the CAD techniques and methods, we modeled all the bracket- and tube-type elements. Using similar CAD techniques, we obtained the final models of the elements, as shown in Figure 12.

To create models of the two orthodontic wires, we scanned the wires in two dimensions using a multifunctional printer. We scaled and these images to a natural scale to take measurements. Then, we uploaded the images to SolidWorks and used them to obtain undefined wire models. In the preliminary orthodontic treatment, we used wires with diameters of 0.014 inches (0.3556 mm). To define models of the orthodontic wires, we uploaded the images to provide a basis for drawing the basic curves that define the shape of the wires. Previously, we scaled and resized the images.

In the defined sketch plan, we drew a Spline curve over the uploaded and scanned image. In the perpendicular plane on this curve, we drew a circle with a diameter of 0.3556 mm, as shown in Figure 13.

We used the two curves (basic curve (Path) and closed curve (Profile)) to define a Sweep-type virtual solid, which is basically the model of the orthodontic wire (Figure 14).

We defined the other orthodontic wire in a similar way. The models of the two orthodontic wires are presented in Figure 15.

As the first step, we placed the bracket and tube elements on the vestibular surfaces of the teeth using CAD methods and techniques in the Assembly module of SolidWorks, as shown in Figure 16.

To obtain models of the deformed orthodontic wires, we defined three points on all the models of brackets and tubes as basic points for defining the Path curves of the wires, as shown in Figure 17.

These three points placed on each bracket and tube element were sufficient to define, in the context of the assembly, the Path base curves of the two orthodontic wires, as shown in Figure 18.

As in the case of undeformed orthodontic wires, we drew circles with diameters of 0.3556 mm (0.014 inches) on planes perpendicular to these curves. Figure 19 presents the model of one orthodontic wire. The model of the other wire was obtained in a similar way.

Once the models of the two wires were generated in the context of the ensemble, the custom model of the orthodontic system was finalized. The final model of the applied orthodontic system is presented in Figure 20.

## 3. Results

The mechanical and elastic properties of materials are of great importance in engineering when evaluating a biomechanical system. These characteristics are determined via mechanical tests carried out in a specialized laboratory, on special machines, or using a virtual laboratory based on finite element analysis software.

Young’s modulus (E), also known as the longitudinal modulus of elasticity, is a value that expresses the stiffness of an elastic and isotropic material. Young’s modulus is defined as the ratio of the axial stress and the axial strain in the range of Hooke’s Law and expressed in N/m^2^.

A stressed solid will deform. This deformation can be elastic, if the body returns to its initial state after cessation of the external force. In the case of an elastic deformation, an elastic force Fe arises inside the deformed body, which opposes the external stress represented by the force F.

If the deformation has the value ∆l, the elastic force has the following expression:Fe = −k ∆l = −F (1)
where k is a coefficient that expresses the elastic constant of the material to be stretched and depends on the material.

In absolute terms, deformation can also be written as
F = k · x(2)
where k is an elastic constant and depends on the material, and x is the displacement obtained under the action of force.

If the analyzed material or element is not metallic, the elastic force can also be expressed by the following relation:F = k · x^n^(3)
where n ≤ 3.

We next applied these theoretical considerations and general principles to the two upper and lower orthodontic wires. To determine a mathematical model of the force caused by the deformation of an orthodontic wire, the Ansys Workbench software, which employs FEM techniques, was used as a test laboratory. Thus, these wires were subjected to forces with different values, and the displacements of the different points on the respective orthodontic wires were measured. Values were recorded using Microsoft Excel. The purpose of these virtual experiments was to determine a functional mathematical model that would allow us to evaluate the elastic force on each bracket- and tube-type element and, subsequently, for these forces to act on the orthodontic system used.

To begin, we analyzed the wire used in the maxillary arcade. Initially, we determined the length of the wire in an undeformed state. This measurement was performed using a virtual model of the wire with virtual measuring instruments. The length of the upper wire was 152.12 mm.

Similarly, we determined the length of the wire used at the level of the mandibular arcade with a length of 143.12 mm. Using the same CAD method of virtual measurement, we set the lengths in the distorted state. We sectioned the wires, adapting their length to the length of the patient’s arches. The length of the upper wire was 109.68 mm, and that of the lower wire was 104.09 mm. Following these determinations, we concluded that the upper wire was shortened by 42.43 mm, and the lower wire was shortened by 39.02 mm.

For each orthodontic wire, we used a special coordinate system. For assembly reasons, we shortened the undeformed wires via CAD techniques using the values set out above to prepare the models for analysis and virtual testing in Ansys Workbench.

In the Engineering Data module of the Ansys Workbench software, we defined the material characteristics of Nitinol, as shown in Table 1.

Next, we set the surfaces considered fixed, specifically the extremities of the orthodontic wire.

In the next step, we divided the orthodontic wire model into finite elements. These finite elements have a tetrahedral shape and a maximum size of 1 mm. In Figure 21, we present images of the upper wire with the structure of finite elements.

Next, we placed a force at a distance z = 10 mm, which corresponded to the coordinates measured on the curve y = 10.001 mm. We realized the first simulation using force F = 0.1 N. Figure 22 presents the position and direction of the force.

After determining these features, we ran the application. Figure 23 presents the resulting displacement map. To determine the exact value of the deformation x, we used the tool called Deformation Probe. In this way, we obtained the value x = 0.4233 mm.

We repeated the simulation for force F values from 0.1 to 10 N and recorded the values obtained in a Microsoft Excel table. We also calculated the value of the elastic constant k according to relation (2)—namely, k = F/x. These values are shown in Table 2.

From an analysis of Table 2, it can be observed that the value of the elastic constant is the same with k = 0.236239 for z = 10 mm, which confirms the equations of the Elasticity Theory. Next, we placed a force at distance z = 20 mm and resumed the simulation for force F = 0.1 N. For this simulation, we obtained the value x = 2.1089 mm. We repeated the simulation for values of force F from 0.1 to 10 N and recorded the obtained values again in a Microsoft Excel table. We also calculated the value of the elastic constant k according to relation (2)—namely, k = F/x. These values are presented in Table 3.

We resumed the simulation for locations of z = 30, z = 40, z = 50, and z = 51.38 mm in the middle of the upper wire. Additionally, force F varied between 0.1 and 10 N. We calculated the value of the elastic constant k according to Relation (2)—namely, k = F/x. Table 4 shows the simulation values for z = 51.38 mm and force F from 0.1 to 10 N.

The obtained results showed that the value of the elastic constant k depends on the position by which the force acts on the orthodontic wire. Figure 24 plots the value of the constant k as a function of the z coordinates.

Clearly, the value of the elastic constant k depends on the position on the orthodontic wire, given either by the linear coordinates z or by the curvilinear coordinates y. In SolidWorks, we measured, on the undeformed upper orthodontic wire, the y coordinates for the values of the z coordinates. We then developed a combined table with the values of both z and y, as well as the values of the constant k (Table 5).

It is also clear that the elastic force developed by the upper wire is dependent on the value of deformation x and the curvilinear y coordinates, so F = F (x, y). Thus, F = k (y) · x. Knowing seven sets of values, we can propose a degree 6 function for the elastic constant k:k = k (y) = a × y^6^ + b × y^5^ + c × y^4^ + d × y^3^ + e × y^2^ + f × y + g. (4)

Using Table 5, we can replace the values for y and k and obtain
a·(0)^6^ + b·(0)^5^ + c·(0)^4^ + d·(0)^3^ + e·(0)^2^ + f·(0) + g = 0
a·(10)^6^ + b·(10)^5^ + c·(10)^4^ + d·(10)^3^ + e·(10)^2^ + f·(10) + g = 0.236239
a·(20.04)^6^ + b·(20.04)^5^ + c·(20.04)^4^ + d·(20.04)^3^ + e·(20.04)^2^ + f·(20.04) + g = 0.004741799
a·(30.23)^6^ + b·(30.23)^5^ + c·(30.23)^4^ + d·(30.23)^3^ + e·(30.23)^2^ + f·(30.23) + g = 0.021268997
a·(40.75)^6^ + b·(40.75)^5^ + c·(40.75)^4^ + d·(40.75)^3^ + e·(40.75)^2^ + f·(40.75) + g = 0.014176989
a·(51.85)^6^ + b·(51.85)^5^ + c·(51.85)^4^ + d·(51.85)^3^ + e·(51.85)^2^ + f·(51.85) + g = 0.014093042
a·(59)^6^ + b·(59)^5^ + c·(59)^4^ + d·(59)^3^ + e·(59)^2^ + f·(59) + g = 0.10383.(5)

We were able to solve the first equation in the system (5) immediately with the solution g = 0.

In this case, Equation (5) forms a system of six equations and six unknowns (a, b, c, d, e, f). Here, the system is determined and the unknowns can be obtained. For this complicated calculation, we used the Maple software, which can solve this system of equations. We obtained the following values for the unknowns:

a = −1.89309451393284 × 10^−9^

b = 3.87447065552008 × 10^−7^

c = −0.305120626818607 × 10^−4^

d = 0.115143861398714 × 10^−2^

e = −0.0207466264631038

f = 0.142773204710058

g = 0

With these known coefficients, we were able to express the force in the upper orthodontic wire as follows:F = (−1.89309451393284 × 10^−9^ × y^6^ + 3.87447065552008 × 10^−7^ × y^5^ −
0.305120626818607 × 10^−4^ × y^4^ + 0.115143861398714 × 10^−2^ × y^3^ −
0.0207466264631038 × y^2^ + 0.142773204710058 × y) × x(6)
where x is the linear deformation due to the elastic force, and y represents the curvilinear coordinates measured at the edges of the wire.

Similarly, we determined the force in the lower orthodontic wire as follows:F = (−1.40083194030139 × 10^−9^ × y^6^ + 3.04238736235189 × 10^−7^ × y^5^ −
−0.254032635123737 × 10^−4^ × y^4^ + 0.102066672881550 × 10^−2^ × y^3^ −
−0.0197861695057057 × y^2^ + 0.149264732878305 × y) × x.(7)

To determine the force values on each bracket and tube element, as seen in the force defining functions, it was necessary to determine the y coordinates and x deformations caused by elastic forces.

To measure the y distances on the deformed wires, we used CAD techniques specific to the SolidWorks software. Figure 25 shows the measurements for 4.5.

Similarly, we determined the y coordinates for all bracket and tube elements and recorded them in Excel tables. In Table 6, we present the y values for the lower wire, and in Table 7, we present the values for the upper wire.

To determine the x deformations, we first overlapped the two models of the wire in undeformed and deformed states.

On these models, we measured the x deformations between similar characteristic points on the two models. For example, Figure 26 presents the measurements of the x deformation for 1.3.

Similarly, we determined all the values for the x deformation.

By applying formulas (6) and (7), we obtained the values of the forces developed in the lower and upper wires, as given in Table 8 and Table 9.

Next, we imported the custom virtual system model into Ansys Workbench. We removed the orthodontic wires from the model, replacing their effects by the action of forces on each bracket- and tube-type element.

In the Engineering Data module of the Ansys Workbench software, we defined the material characteristics of the components of the analyzed system, as presented in Table 10.

We divided the model into tetrahedral finite elements with a maximum size of 20 mm. Next, we indicated the elements considered fixed (maxillary and mandible). Then, we set the forces and values (from Table 8 and Table 9) for each element, as shown in Figure 27.

After running the application, we obtained maps of displacement (Figure 28), deformation (Figure 29), stress (Figure 30), and deformation energy (Figure 31).

## 4. Discussion

Malocclusions are generally treated using three-dimensional orthodontic forces. However, in the past, the diagnosis of malocclusions was established via two-dimensional imaging investigations. Although these techniques also provide useful information, to ensure the accuracy of an orthodontic diagnosis, it is necessary to also include three-dimensional imaging investigations, the most commonly used of which is the CBCT [13].

In this study, using the CBCT technique, we obtained detailed images of the structures of the maxillary and dental bones, which we transferred to the InVesalius program in order to achieve a three-dimensional reconstruction. Subsequently, we processed the anatomical geometries using the Geomagic program, resulting in a model that we transformed into a virtual solid using the SolidWorks program. The final analysis using the FEM technique was performed using the Ansys Workbench program. In this way, we obtained maps of results that helped us establish the mechanical behavior of the orthodontic system.

In the literature, there are few studies on the modeling and simulation of an orthodontic system using virtual methods.

In orthodontics, linear measurements are commonly performed at the level of a virtual model based on CBCT imaging investigations. These investigations have been shown to be much more effective than analyses using conventional study models resulting from alginate or silicone impressions. However, the virtual models were obtained from a normal CBCT scan, with the patient in occlusion. Thus, there was an overlap of the upper and lower occlusal surfaces. Conventional study models record the occlusal surfaces as a whole, enabling the morphology to be better reproduced. Taking into account the characteristics of the two types of models with which linear measurements were created, further studies will be needed to determine the accuracy of the methods [14].

Other studies highlight the importance of CBCT techniques in orthognathic surgery. The full preparation of patients for such surgery is performed digitally, using three-dimensional images captured via CBCT and special software that simulates the final result of treatment. The use of a virtual model facilitates the work of the medical team, suggesting the appropriate types therapeutic displacement that should be performed in the surgical phase and orthodontic phase. Postoperatively, the model can also assess whether surgical treatment was performed according to virtual planning [15,16,17].

In terms of using FEM to simulate various clinical scenarios, most researchers argue that the optimal force exerted during orthodontic treatment should be up to 1 N. This may be true in theory, but in practice, it is complicated to control and evaluate the intensity of the force applied to each tooth. When the force has higher value, there is a risk of side effects such as bone and root resorption, increased tooth mobility, patient discomfort, etc. [18,19,20,21].

A study conducted using FEM analyzed the stress distribution in the periodontal ligament of the central incisor according to the thickness of the transparent aligner during orthodontic treatment. The authors concluded that the principal stresses induced by the aligner in the periodontal ligament of the central incisor were within ranges sufficient to induce remodeling of the periodontal ligament to produce sufficient tooth movement [22].

Based on the results of another FEM analysis, the authors confirmed that overhanging attachments (OA) can control the orthodontist’s unintentional tooth movement better than general attachments (GA). OA are considered to reduce the risk of the attachment becoming detached during orthodontic treatment by highlighting desirable stress distributions and reducing the stress concentration between the attachment and the aligner [23].

## 5. Conclusions

Analyzing the result maps obtained from the study, we concluded that the maximum mechanical displacements were 9.9617 × 10^−7^ m (0.996 microns), particularly on the incisors. The maximum deformations were 0.00015801 and were found, in particular, on the contact surfaces between the bracket- or tube-type elements and the teeth. The maximum stress was 1.0595 × 10^7^ Pa and observed on the incisors and the contact surfaces between the brackets and the teeth. The deformation energy was 1.4487 × 10^−8^ J and found, in particular, on the incisors. However, further experimental studies are needed to deepen the results obtained.

## Figures and Tables

**Figure 1 diagnostics-12-01296-f001:**
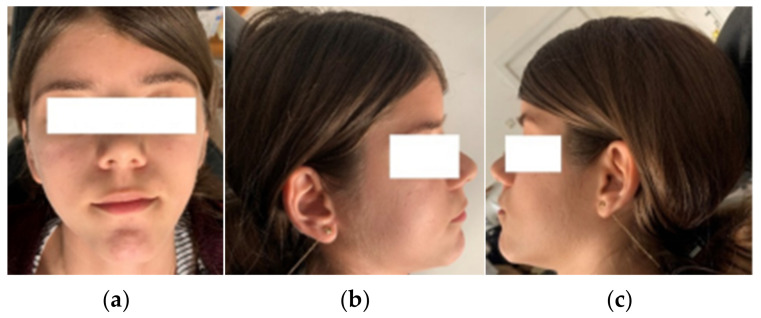
Front (**a**), right (**b**), and left (**c**) profile photos of the patient.

**Figure 2 diagnostics-12-01296-f002:**
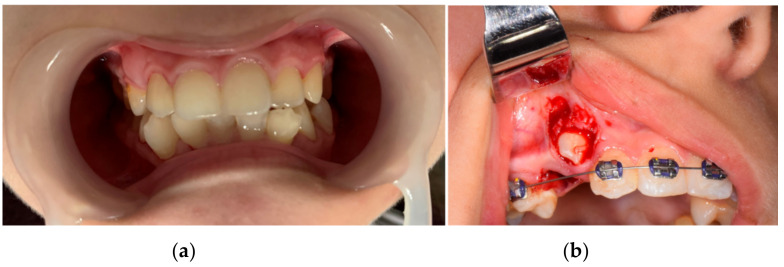
Intraoral photos in occlusion (**a**) and from the surgical intervention (**b**).

**Figure 3 diagnostics-12-01296-f003:**
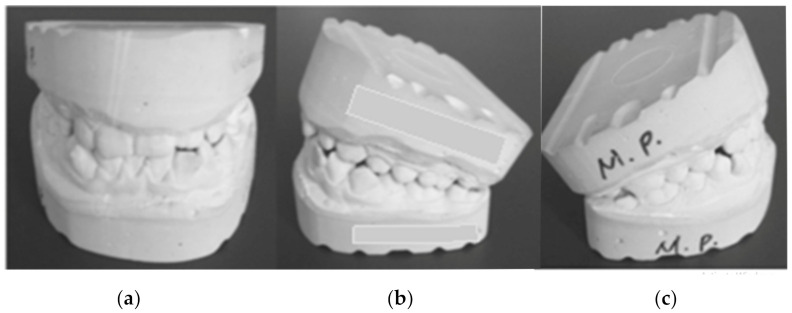
Patient’s plaster study models: (**a**) front image, (**b**) left-side image, (**c**) right-side image.

**Figure 4 diagnostics-12-01296-f004:**
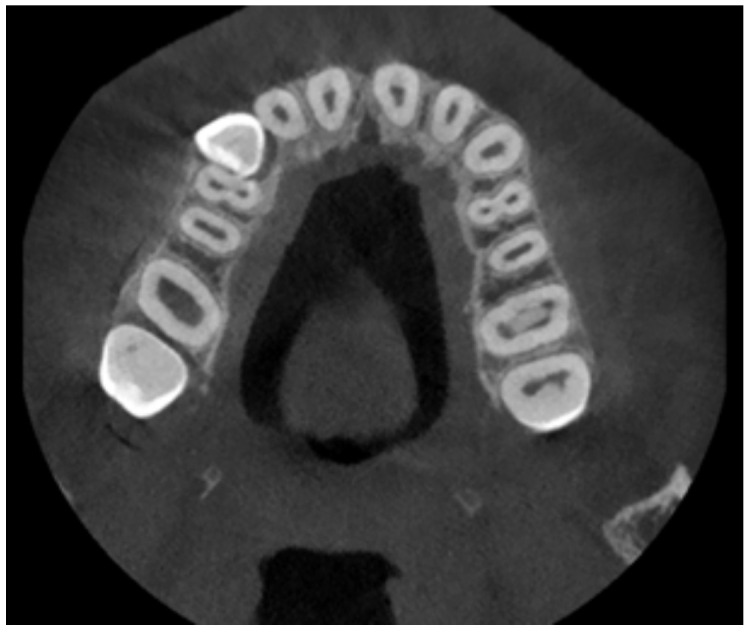
A tomographic image of the analyzed patient.

**Figure 5 diagnostics-12-01296-f005:**
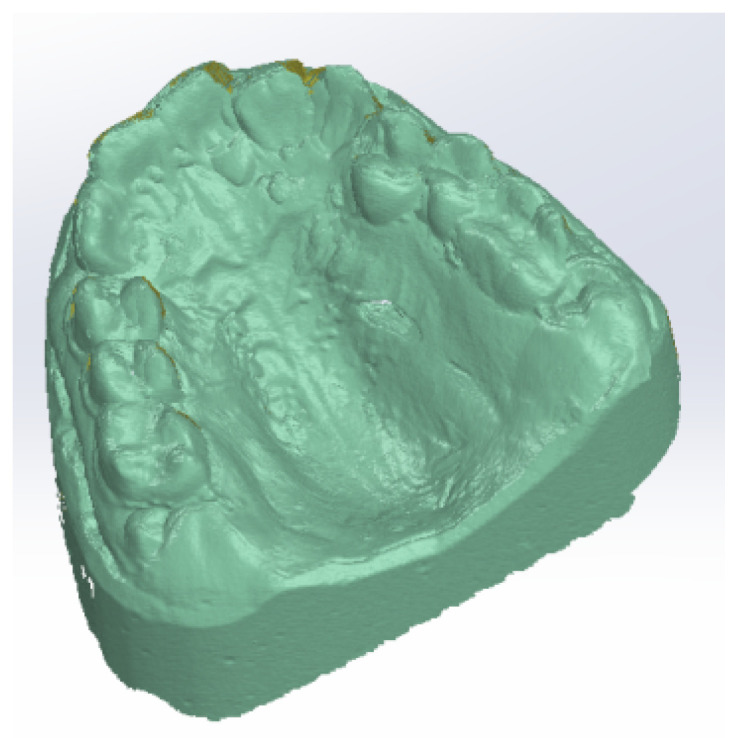
Scanned image of the maxillary arch.

**Figure 6 diagnostics-12-01296-f006:**
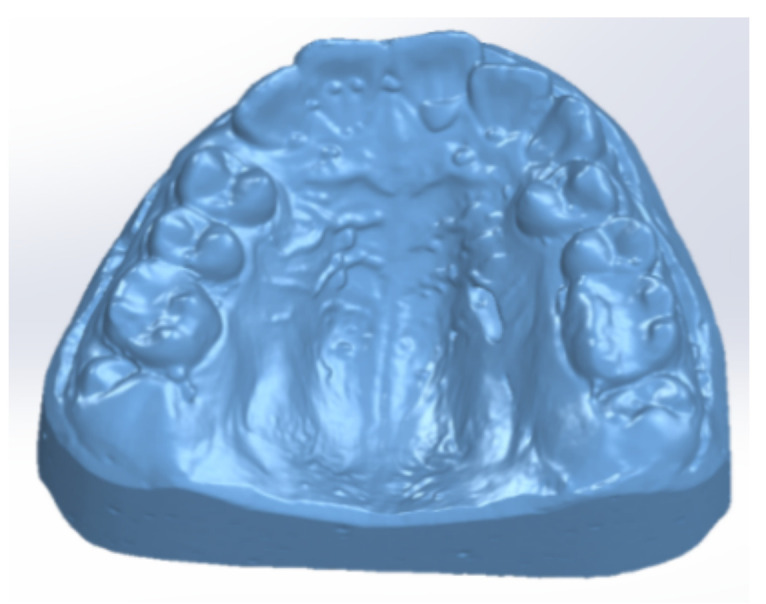
Perfectly closed surface obtained after the filling operation.

**Figure 7 diagnostics-12-01296-f007:**
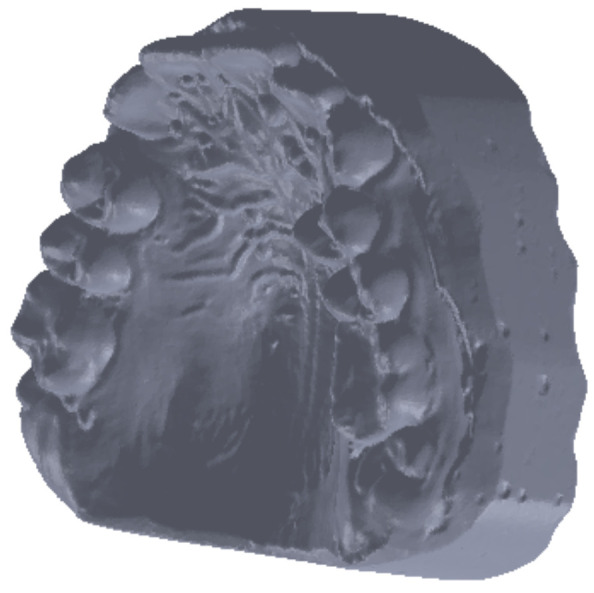
The final virtual solid of the scanned maxillary model.

**Figure 8 diagnostics-12-01296-f008:**
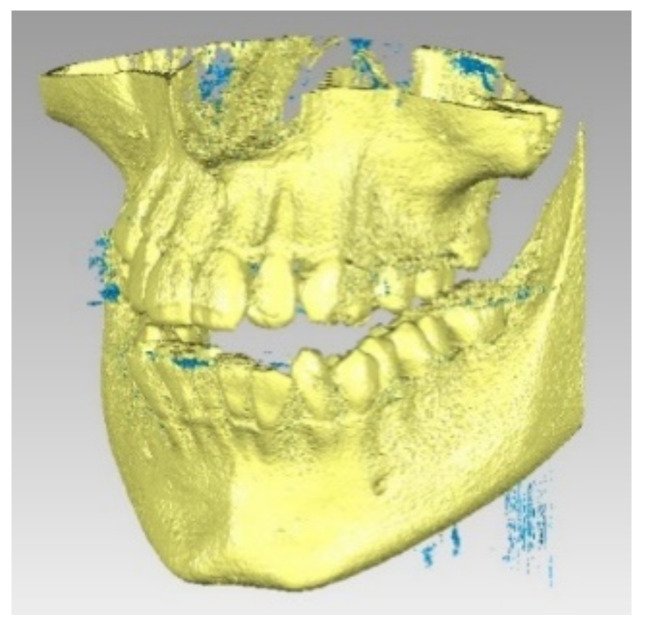
The primary bone and dental geometry of the patient studied in the Geomagic software.

**Figure 9 diagnostics-12-01296-f009:**
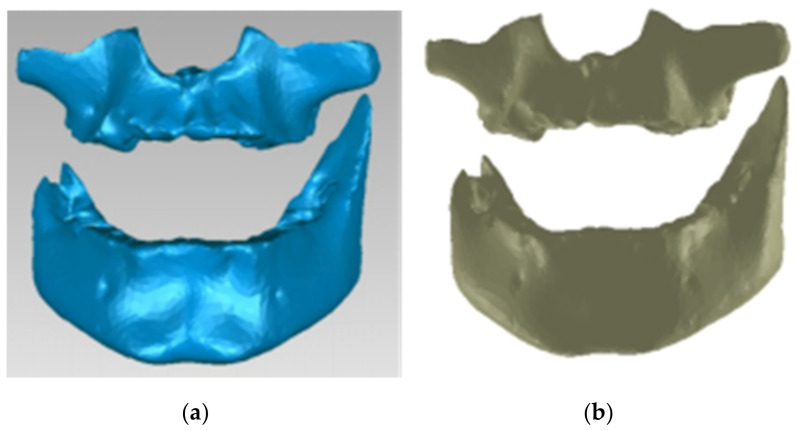
The final model of osseous components in Geomagic (**a**) and SolidWorks (**b**).

**Figure 10 diagnostics-12-01296-f010:**
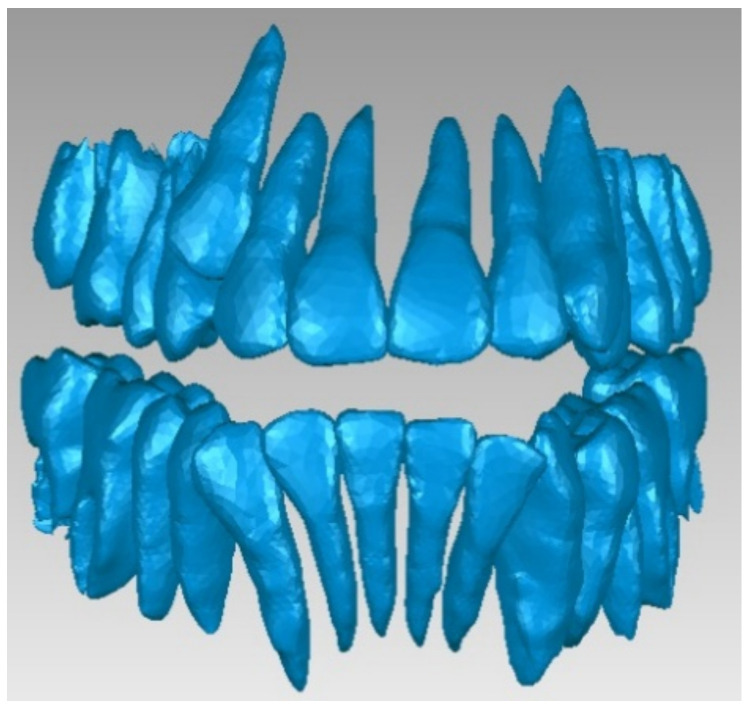
The final dental structure in Geomagic.

**Figure 11 diagnostics-12-01296-f011:**
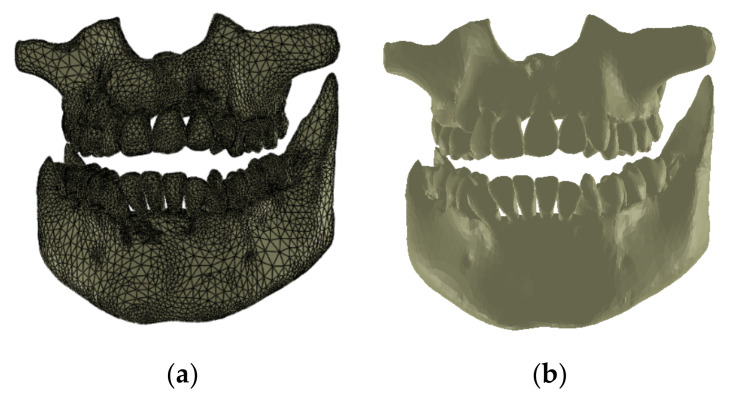
Complete structure—two viewing modes (**a**,**b**).

**Figure 12 diagnostics-12-01296-f012:**
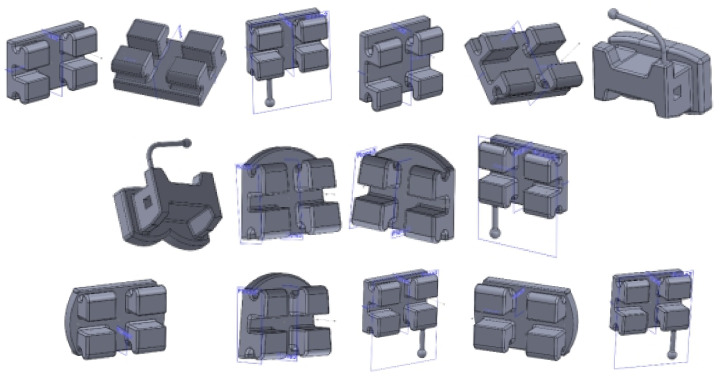
Virtual models of bracket and tube elements.

**Figure 13 diagnostics-12-01296-f013:**
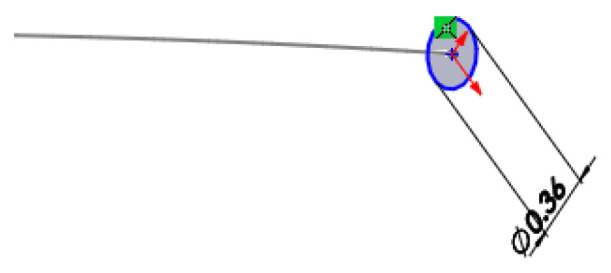
Drawing a circle in a perpendicular plane to the Spline curve.

**Figure 14 diagnostics-12-01296-f014:**
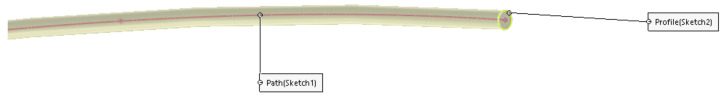
Defining the sweep shape.

**Figure 15 diagnostics-12-01296-f015:**
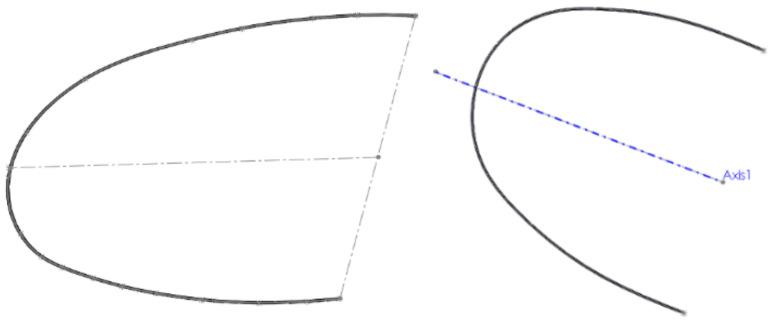
Virtual models of the two undeformed orthodontic wires.

**Figure 16 diagnostics-12-01296-f016:**
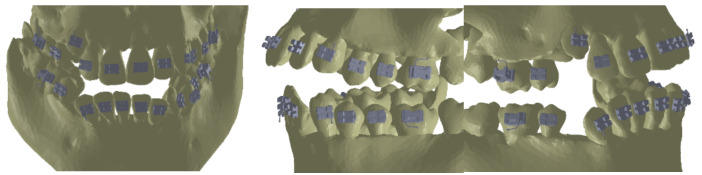
Location of the bracket and tube elements.

**Figure 17 diagnostics-12-01296-f017:**
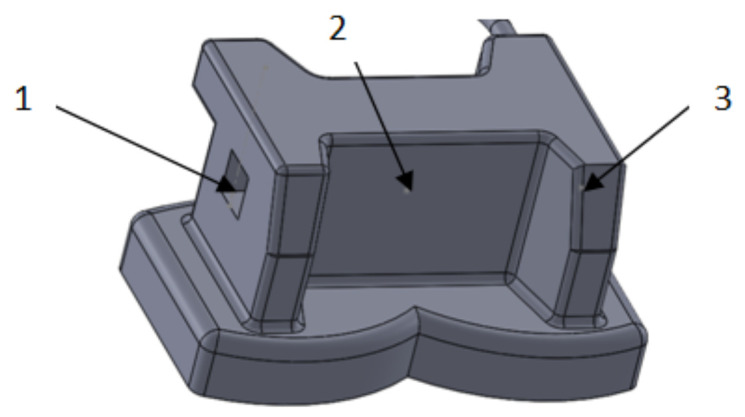
Three guide points attached to a tube.

**Figure 18 diagnostics-12-01296-f018:**
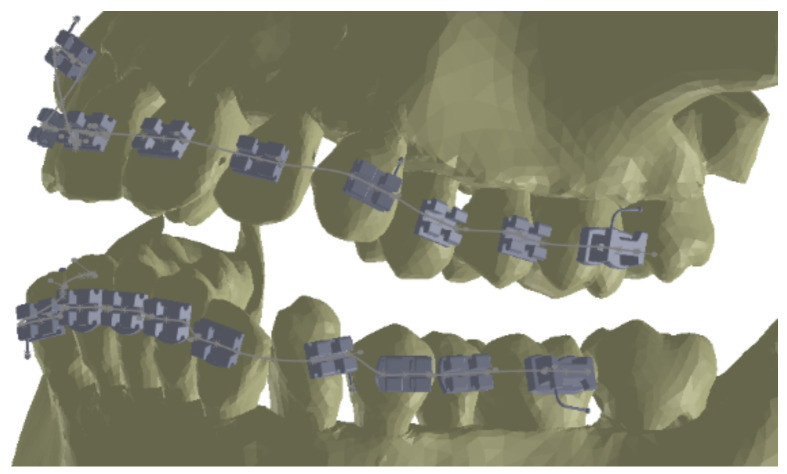
Path base curves generated for the two orthodontic wires.

**Figure 19 diagnostics-12-01296-f019:**
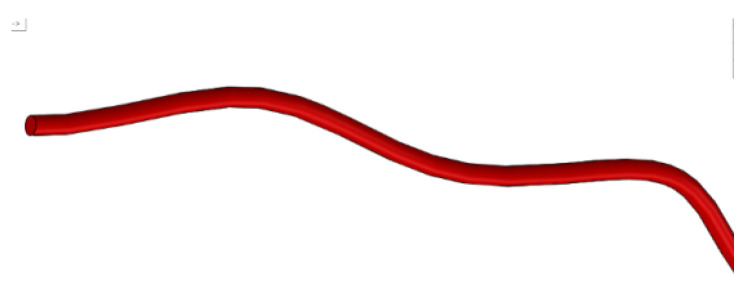
Model of a deformed orthodontic wire.

**Figure 20 diagnostics-12-01296-f020:**
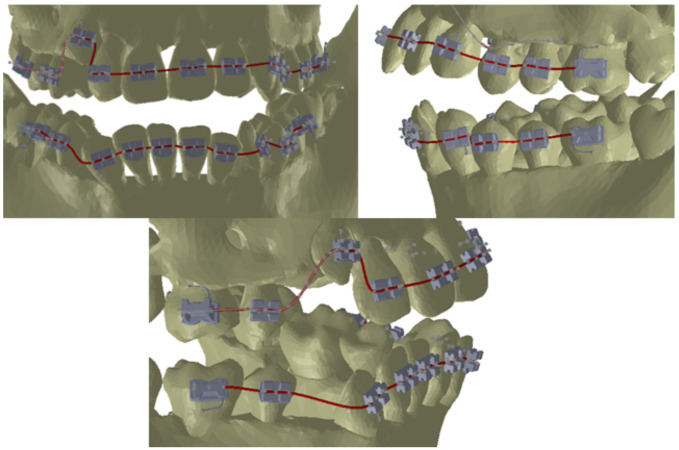
Model of the orthodontic system.

**Figure 21 diagnostics-12-01296-f021:**
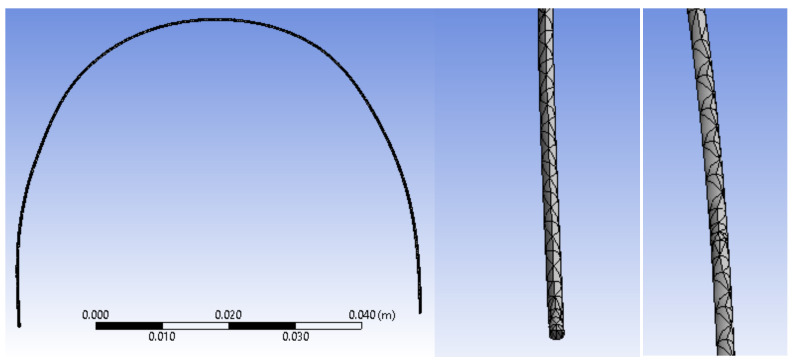
The finite element structure of the upper wire model.

**Figure 22 diagnostics-12-01296-f022:**
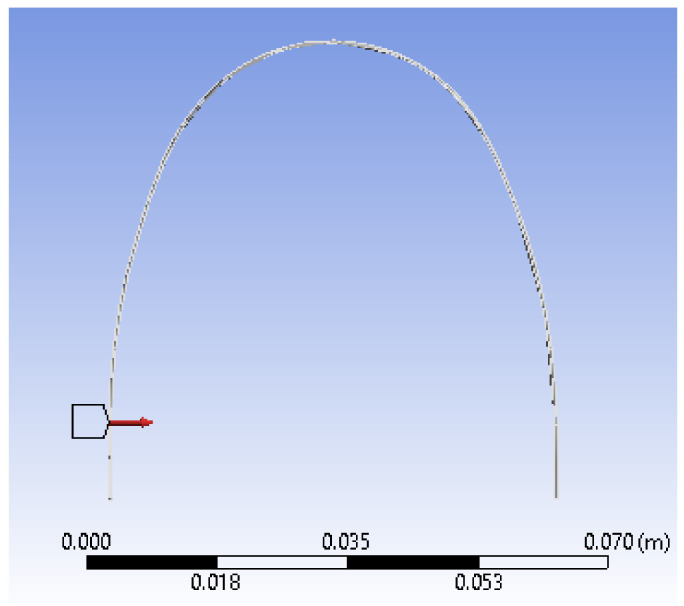
Position and direction of the force.

**Figure 23 diagnostics-12-01296-f023:**
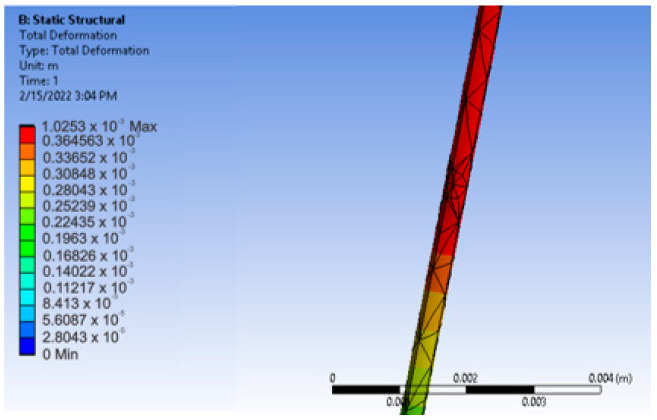
Displacement map for z = 10 mm and force F = 0.1 N.

**Figure 24 diagnostics-12-01296-f024:**
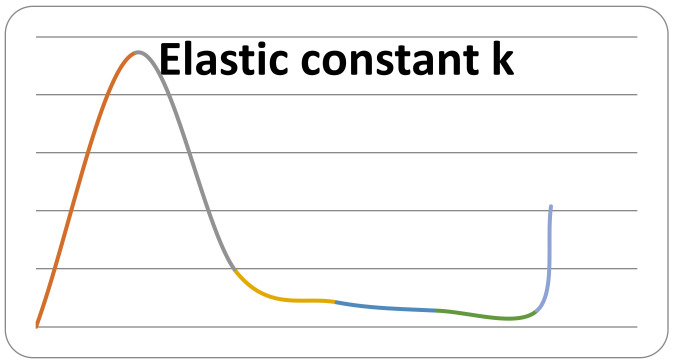
Value of constant k as a function of the z coordinates for the upper wire.

**Figure 25 diagnostics-12-01296-f025:**
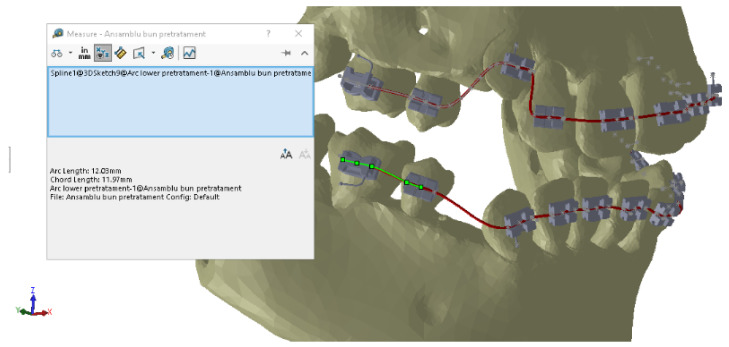
Measurements of the y coordinates for 4.5.

**Figure 26 diagnostics-12-01296-f026:**
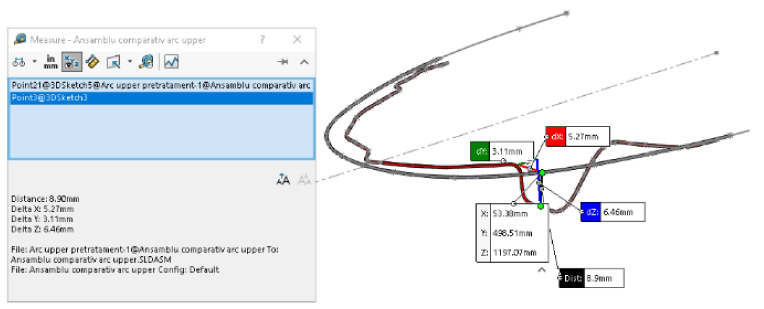
Measurement of the x deformation for 1.3.

**Figure 27 diagnostics-12-01296-f027:**
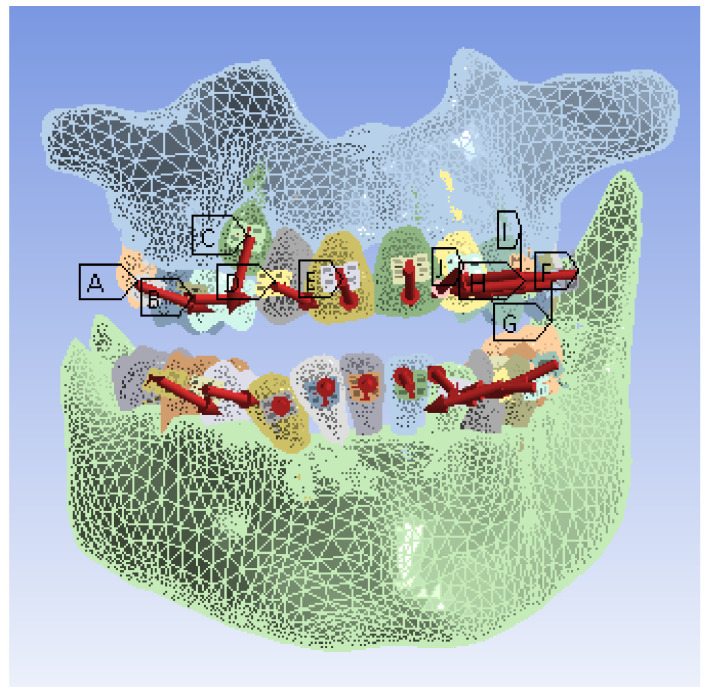
The forces defined in Ansys for the analyzed model.

**Figure 28 diagnostics-12-01296-f028:**
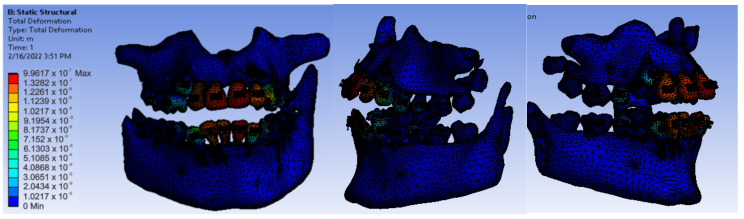
Displacement maps.

**Figure 29 diagnostics-12-01296-f029:**
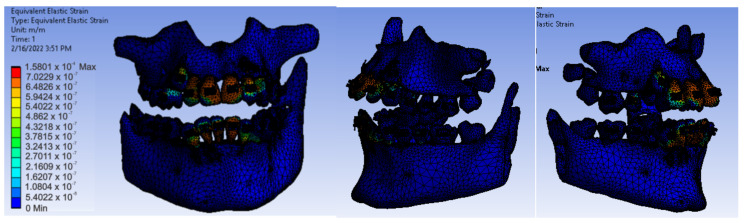
Deformation maps.

**Figure 30 diagnostics-12-01296-f030:**
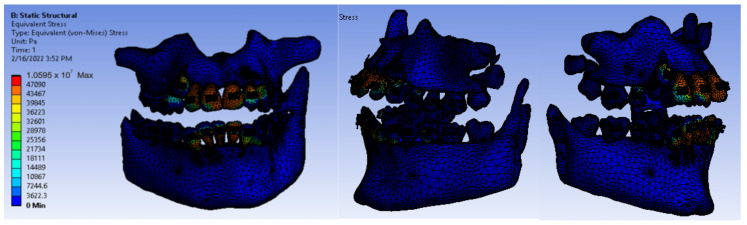
Stress maps.

**Figure 31 diagnostics-12-01296-f031:**
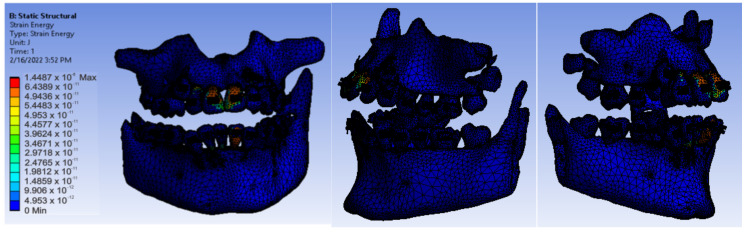
Deformation energy maps.

**Table 1 diagnostics-12-01296-t001:** Physical properties of Nitinol used in simulations.

Material	Density	Young’s Modulus	Transverse Modulus of Elasticity	Poisson’s Coefficient
Nitinol	6450 Kg/m^3^	8.3 × 10^7^ Pa	3.12 × 10^7^ Pa	0.33

**Table 2 diagnostics-12-01296-t002:** Simulation values for z = 10 mm and force F from 0.1 to 10 N.

Z = 10 mm		
F [N]	x [mm]	Calculated k
0.1	0.4233	0.236239
0.2	0.8466	0.236239
0.3	1.2699	0.236239
0.4	1.6932	0.236239
0.5	2.1165	0.236239
0.8	3.3864	0.236239
0.9	3.8097	0.236239
1	4.233	0.236239
2	8.466	0.236239
5	21.165	0.236239
10	42.33	0.236239

**Table 3 diagnostics-12-01296-t003:** Simulation values for z = 20 mm and force F from 0.1 to 10 N.

z = 20 mm		
F [N]	X [mm]	Calculated k
0.1	2.1089	0.047418
0.2	4.2178	0.047418
0.3	6.3267	0.047418
0.4	8.4356	0.047418
0.5	10.5445	0.047418
0.6	12.6534	0.047418
0.7	14.7623	0.047418
0.8	16.8712	0.047418
0.9	18.9801	0.047418
1	21.089	0.047418
2	42.178	0.047418
3	63.267	0.047418
4	84.356	0.047418
5	105.445	0.047418
10	210.89	0.047418

**Table 4 diagnostics-12-01296-t004:** Simulation values for z = 51.38 mm (middle of upper wire) and force F from 0.1 to 10 N.

z = 51.38 mm (Middle of the Upper Wire)		
F [N]	x [mm]	Calculated k
0.1	0.96311	0.10383
0.2	1.92622	0.10383
0.3	2.88933	0.10383
0.4	3.85244	0.10383
0.5	4.8156	0.103829
0.6	5.77872	0.103829
0.7	6.74184	0.103829
0.8	7.70496	0.103829
0.9	8.66808	0.103829
1	9.6311	0.10383
2	19.2622	0.10383
3	28.8933	0.10383
4	38.5244	0.10383
5	48.1555	0.10383
10	96.311	0.10383

**Table 5 diagnostics-12-01296-t005:** The values of the elastic constant k as a function of the z and y coordinates.

Y	Z	K	k (y)
0	0	0	0
10	10	0.236239074	0.236239
20.04	20	0.047418085	0.004741799
30.23	30	0.021269355	0.021268997
40.75	40	0.014176957	0.014176989
51.85	50	0.014092645	0.014093042
59	51.38	0.1038303	0.10383

**Table 6 diagnostics-12-01296-t006:** Y coordinate values for the lower wire.

Teeth	Measured y
4.6	28.65
4.5	38.32
4.3	54.41
4.2	61.03
4.1	66.41
3.6	29.09
3.5	38.23
3.4	47.03
3.3	54.81
3.2	65.17
3.1	71.25

**Table 7 diagnostics-12-01296-t007:** Y coordinate values for the upper wire.

Teeth	Measured y
1.6	25.66
1.5	35.71
1.3	52.63
1.2	63.43
1.1	73.11
2.6	25.66
2.5	34.83
2.4	43.06
2.3	51.51
2.2	61.49
2.1	70.36

**Table 8 diagnostics-12-01296-t008:** The x and y coordinate values for the lower wire.

Teeth	Measured y (mm)	Measured x (mm)	Elastic Force Developed
4.6	28.65	1.31	0.017648
4.5	38.32	2.75	0.069992
4.3	54.41	3.07	0.140214
4.2	61.03	4.45	0.461278
4.1	66.41	5.41	0.961619
3.6	29.09	1.22	0.019191
3.5	38.23	3.08	0.079447
3.4	47.03	1.11	0.01033
3.3	54.81	3.24	0.166461
3.2	65.17	3.13	0.178164
3.1	71.25	4.74	5.906798

**Table 9 diagnostics-12-01296-t009:** The x and y coordinate values for the upper wire.

Teeth	Measured y (mm)	Measured x (mm)	Elastic Force Developed
1.6	25.66	2.08	0.002101
1.5	35.71	4.56	0.146577
1.3	52.63	9.2	0.207517
1.2	63.43	8.15	0.384285
1.1	73.11	5.84	11.8842
2.6	25.66	1.6	0.001616
2.5	34.83	2.29	0.074832
2.4	43.06	4.28	0.009652
2.3	51.51	2.62	0.028262
2.2	61.49	4.81	0.474462
2.1	70.36	4.65	4.449571

**Table 10 diagnostics-12-01296-t010:** Physical properties of materials used in the simulations.

Component	Material	Density	Young’s Modulus	Transverse Modulus of Elasticity	Poisson’s Coefficient
Bracket- and tube-type elements	Ni + Cr alloy	8500 Kg/m^3^	2.1 × 10^11^ Pa	8.015 × 10^10^ Pa	0.31
Maxillary, mandible	Bone	1400 Kg/m^3^	1 × 10^10^ Pa	3.84 × 10^9^ Pa	0.31
Teeth	Enamel	2958 Kg/m^3^	7.79 × 10^10^ Pa	2.996 × 10^10^ Pa	0.3

## Data Availability

The authors declare that the data from this research are available from the corresponding authors upon reasonable request.

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
