# Peer review of "Modeling and Simulating an Orthodontic System Using Virtual Methods"

_diagnostics, 2022, doi:10.3390/diagnostics12051296_

Round 1
Reviewer 1 Report
Dear Authors,
you made a really great work!

Author Response
Thank you for reviewing our article. Following your suggestions, we've been able to improve the content of this article. Thank you!

Reviewer 2 Report
Manuscript ID: diagnostics-1698952
Title: Modeling and simulating an orthodontic system using virtual methods
1.What is the main question addressed by the research?
To determine a functional mathematical model, which would allow the evaluation of the elastic forces intensity on each bracket and tube type element and the way in which they act on the orthodontic system used.
2.Is it relevant and interesting?
The article is relevant and interesting.
3.How original is the topic?
The topic is current.
4.What does it add to the subject area compared with other published material?
The authors have collected and analyzed a great deal of original data.
5.Is the paper well written?
Yes, the article is well written.
6.Is the text clear and easy to read?
Yes, but moderate English editing is required.
7.Are the conclusions consistent with the evidence and arguments presented?
Yes, the conclusions consistent with the evidence and arguments presented but further studies are needed.
8.Do they address the main question posed?
Yes, the Authors addressed the main question posed.
Other comments:
- English language: moderate English editing is required
- Summary of abbreviations required.
- Introduction: This section needs few improvements. For example, Authors may include a brief sentence on the effect of etiological factors on craniofacial development based on the following reference: <<Several etiological factors contribute to impaired craniofacial development and, in some cases, there is a need to correct these also through orthodontic treatment [https://doi.org/10.3390/jcm10184057]>>.
- Methods: This section has been properly prepared.
- Results: This section has been properly prepared.
- Discussion: What is the main theme that emerges from the authors' analysis?Is the studies design a limitation for this review? Please discuss also the results of these papers: https://doi.org/10.3390/ma14020324 ; https://doi.org/10.3390/ma14174926.
- Conclusion: This section has been properly prepared.
After making the indicated changes, the article may be suitable for publication.
Thanks for the opportunity to review this manuscript.
Author Response
Thank you for your review. Thank you for your help in optimizing our article. Therefore, we've made the changes you suggested. We will also use MDPI’s English language editing services.

Reviewer 3 Report
- First of all, the article should be rewritten by an english speakerscientific
-Title: don't use capital letters for all the words
-Abstract: please clarify what is CBCT (use CBCT as an abbreviation after Cone beam....)
Introduction
- Line 49: [1-3] please correct all the references in the text- The introduction is very poor, where is the aim of the study, the null hypothesis? what is the originality of the present study?
Materials and methods
-Please mention the figure directly after the concerned sentence
- Please when there is a figure (1, 2 and 4) with multipanels, please use a, b, c...... and mention it in the legend
- Figure 2: higher quality is recommended
- Figure 3: higher quality is recommended
- Line 94-95: please clarify or rephrase the sentence
- Line 116: it is specialized..... please clarify?
- 129-131: for simple and for complicated calculations: the authors should explain when it is simple and the complexity
- 132-134: any references for these methods?
- Figure 5: the same for figures 1, 2 and 4; and, please use arrows for the point cloud
- Figure 6: same for figure 5
- The authors could make a figure with multipanels from Figure 5, 6 and 7
- 167-168: it's a scientific article, please revise the sentence
- 178-179: how .stl file was exported?
- Figure 8: very difficult to investigate, please clarify
- Figure 9: higher quality is recommended
- Figure 11: the same for figure 5
- Discussion should include the comparison with other studies on the same subjects
- Conclusions: very long sentences
- References should be formatted following MDPI style, specially 9-12
Author Response
Thank you for agreeing to be our reviewer. Thank you for the timely solutions you have provided. We have modified the article according to your requirements. We will also use MDPI’s English language editing services.

Reviewer 4 Report
This paper is actually a case presentation. Authors mentioned that a cbct examination has been performed. It is unclear which selection criteria have been followed and it is obvious that the ALARA or ALADA principle has not been taken in consideration at all, since it seems that a 2D imaging method should have been appropriate. Moreover, the difference of the clinical results of this method vs the traditional methods is not clear.Is the success rate of the case different or what have been achieved with the traditional treatment planning?
Author Response
Thank you for your time, but your requests are not related to our research.
Round 2
Reviewer 3 Report
The manuscript is well revised based on the suggestions and comments.
Author Response
Good morning! We have taken note of your requirements regarding our article. We have made all the changes you requested and yet your requirements are the same. The citations requested by you are already made and the English language editing was done at the MDPI service. Thank you!

Reviewer 4 Report
According to their response, unfortunately, the authors did not consider my comments and did not bother to answer.
As I have already mentioned according to my opinion this is a single case study and not a research study since it is based on one case. Although the authors did not accept that It seems that they agreed since in the conclusion section they mentioned that "further experimental studies are needed to deepen the results obtained".
Moreover, in the discussion section the authors mentioned "Although these techniques also provide useful information, to ensure the accuracy of an orthodontic diagnosis, it is necessary to also include three-dimensional imaging investigations, the most commonly used of which is the CBCT [13]". According to the official publication of the European Commission, CONE BEAM CT FOR DENTAL AND MAXILLOFACIAL RADIOLOGY Evidence-Based Guidelines (available https://www.sedentexct.eu/ files/ radiation_ protection_172.pdf) to ensure the accuracy of an orthodontic diagnosis, it is not necessary to include three-dimensional imaging investigations, it is important to apply selection criteria to each case. So, I am still wondering was it necessary to perform the CBCT scanning for this case or it was performed for the purpose of this project?
Author Response

(The authors gave the same response as above.)
